# Quantifying Uncertainties in Nonlinear Dynamics of a Modular Assembly Using the Resonance Decay Method

Chengrong Lin [†], Ziheng Zhao [†], Zhenyu Wang, Jianping Jiang, Zhigang Wu and Xing Wang *

School of Aeronautics and Astronautics, Shenzhen Campus of Sun Yat-sen University, Shenzhen 518107, China
* Correspondence: wangxing5@mail.sysu.edu.cn
† These authors contributed equally to this work.

**Abstract:** Modular assembling is a promising approach to constructing large spacecraft beyond the size limitations posed by launch vehicles. However, the uncertainties and nonlinearities of the dynamics associated with the assembled structure are deeply concerned with the design stage of such a spacecraft. Conventionally, this concern can be relieved by performing Ground Vibration Testing (GVT) of the structure. Nevertheless, it is challenging for a modular assembly, in which a very low-frequency behaviour and a lack of dynamic testing procedure that can incorporate nonlinearities are two major obstacles. In this regard, the present paper first introduces a demonstrator of Large Structure Assembly (LSA demonstrator), which includes a soft-bungee suspension system, a 6 m long modular assembly, a vibration control system, and a noncontact measurement system. Secondly, a new quantification procedure for the modular assembly, which utilises the resonance decay method, was proposed in this paper. Detailed test steps were illustrated through the demonstrator, in which the backbone curves were treated as key measurement targets in quantifying its nonlinear dynamics. The uncertainties in nonlinear dynamics were also evaluated by assembling and disassembling the structure multiple times. Results have shown that the proposed procedure can efficiently and accurately quantify the dynamics of a highly flexible, large-scale modular assembly.

**Keywords:** uncertainty quantification; large structure assembly; resonance decay method



## 1. Introduction

The need for larger spacecraft has been continuously pushed by greater scientific exploration ambitions such as astrophysics observatories [1], persistent asserts [2], and solar power satellites [3]. The dimensions of these proposed spacecraft reach hundreds or even thousands of meters, which are too large to package into a single launch vehicle. To shorten its development lifecycle and thereby reduce construction costs, the in-space assembly concept was proposed to improve the spacecraft's size [4]. As early as the 1970s, NASA conducted a series of studies into designing and constructing space structures larger than the then-available and proposed launch vehicles. In one of the on-orbit extra-vehicular activities (EVAs) of Orbiter Atlantis, the Assembly Concept for Construction of Erectable Space Structure (ACCESS) experiment was designed to explore the feasibility of manual assembly. The task was to assemble a 13.7 m long truss structure by two astronauts working in space suits in the space shuttle cargo bay. Results of the ACCESS experiment showed that the in-space construction time for truss structures could be reliably estimated in a simulated zero-G (underwater) environment on the ground [5]. Subsequently, many more construction tasks were investigated in a simulated environment. Therefore, the efficiency and reliability of the assembling procedures can be carefully evaluated a priori. For example, a concept for the assembly of the keel structure of the Space Station Freedom (SSF) truss (110 m in length with a 5 m square cross section) using a mobile transporter was evaluated in neutral buoyancy tests [6]. In another example, a precision reflector with a 14 m diameter was constructed in simulated EVAs at NASA's Langley Research

Center (LaRC) [7]. Almost in parallel with the studies on manual assembly, the automated robotic assembly of an 8 m tetrahedral truss [8] and a truss beam [9] were successfully demonstrated. The associated technologies, such as executive control systems, machine vision, automated sequence planning, and automated path planning, were also validated in ground tests [9].

Different from the abovementioned historical approaches of assembling space structures using basic structural elements (e.g., struts, panels, and connecting nodes), a new architecture called modular assembly [2] has recently drawn much attention. It considers assembling a large or ultra large space structure through modular structural units. In LaRC, the construction of a generic persistent platform using the modular approach was studied in detail. The backbone truss structure of the spacecraft was proposed to be assembled from modules using a long-reach manipulator. It also reserves interfaces for adding additional solar tug modules or new science modules [2]. An evolvable space telescope is another type of target spacecraft proposed to be constructed using the modular approach. Its primary mirror can be assembled using a series of tri-truss [2] or hexagonal truss modules [10]. This new modular approach is believed to own many advantages: (1) each modular structural unit can be integrated, verified, and characterised prior to launch; (2) the whole spacecraft can still be upgraded or repaired at relatively low costs; (3) the structural dynamics of the assembly are claimed to be more predictable and repeatable compared with those assembled using conventional approaches [2], despite this claim not being well addressed in the literature.

This paper focuses on quantifying the structural dynamics of a modular assembly through GVT. From the viewpoint of structural dynamics, the lightweight, deployable module is highly flexible, featuring very low-frequency and closely spaced modes [11]. As such, it may vibrate at large amplitudes when subjected to external forces, making it susceptible to geometrical nonlinearities. In another aspect, the modular interface has discontinuous stiffness or even clearance due to manufacturing tolerances, leading to micro- or macroslips around localised areas as the assembly undergoes vibrations; this is often referred to as localised nonlinearity in the structural dynamics community. The geometrical or localised nonlinearities then contribute to the overall nonlinear dynamic behaviours of the modular assembly. In addition to all these, the modular interfaces are almost guaranteed to encounter many uncertain factors, such as variability of contact loads of the interfacing mechanisms and uncertain frictional forces on clamping surfaces [12]. Therefore, the uncertainties and nonlinearities of the dynamics of a constructed modular assembly are of great concern in its design stage and must be carefully quantified in ground vibration tests.

However, conventional GVT procedures of spacecraft structures rely on linear and deterministic assumptions of structural models [13]. The techniques in current practice suffer from significant difficulties in dealing with nonlinear and uncertain modular assemblies in many aspects. First, the vibrations of a modular assembly typically belong to a very low-frequency regime (e.g., below 3 Hz), which is lower than the dynamic range of most conventional piezoelectric accelerometers. In this regard, new measurement systems with improved capabilities in low-frequency ranges need to be explored and validated. Second, the frequency response functions (FRFs), commonly used as measurement targets in experimental modal analysis, are amplitude-dependent for a nonlinear structure [14,15]. To obtain these FRFs, multiple tests at various input levels have to be used, requiring a prohibitive amount of time for the testing of highly flexible structures. Third, a modular assembly also requires a dedicated gravity offloading system to provide free–free boundary conditions [16] and to mitigate the gravity loading effects on the modular interfaces. Because of the abovementioned challenges, experimental quantification of the nonlinearities and uncertainties of a modular assembly has not been reported in the available literature to the authors' knowledge.

To address these challenges, this paper introduces a demonstrator of Large Structure Assembly (LSA demonstrator) developed at Sun Yat-sen University, aiming to investigate

advanced vibration testing techniques for large-scale modular assemblies. The demonstrator consists of a soft-bungee suspension system, three mock-up modular structural units assembled using quick-attachment quick-release joints, a vibration control system to apply excitations, and a noncontact motion capture system (NOKOV) for 3D dynamic displacement measurement. Subsequently, a new quantification procedure for structural nonlinearities and uncertainties was proposed and applied to the LSA demonstrator: (1) The underlying linear modal parameters of the test structure were first identified using impact testing and operational modal analysis. (2) The nonlinear dynamics of a mode were quantified using backbone curves, which were experimentally estimated using the resonance decay method. (3) The structural dynamics' uncertainties were then quantified by assembling and disassembling the structure several times, and a backbone curve was estimated for each assembled structure and each of its modes.

The following paper is organised as follows. Section 2 describes the LSA demonstrator and its four subsystems in detail. Then, Section 3 presents the detailed quantification steps with applications to the LSA demonstrator, where linear and nonlinear dynamic testing results are elaborated. Finally, conclusions are drawn in Section 4.

## 2. Ground Demonstrator of Large Structure Assembly

In this section, the LSA demonstrator, developed by the dynamics group at Sun Yat-sen University, is introduced. Then, four of its subsystems—the suspension setup, the modular units and joints, the vibration control system, and the measurement system—are described in detail, including the context behind their design and manufacture.

### 2.1. The Suspension Setup

In a GVT campaign, the test structure is often suspended or supported by a soft-bungee system to simulate free–free boundary conditions. One challenge in the GVT for a highly flexible assembly is designing a suspension setup with enough strength to support the test structure but also feature a desired ultralow stiffness to avoid coupling to its flexible dynamics [17]. Figure 1 shows the setup of the LSA demonstrator, where an aluminium profile frame was built as the main support structure, measuring 6 m in length, 2 m in width, and 3 m in height. The stiffness of the frame was enhanced by multiple crossbars and columns. As shown in the figure, two ultrasoft bungees were used to suspend each modular unit. These bungees were soft enough so that the rigid body modes of the module were separated, as much as possible, from its flexible modes of interest. Another challenge in the GVT of an assembly is that each modular unit needs to be aligned horizontally to avoid gravitational loading to the modular interfaces; this was met by connecting one end of the bungee to a self-locking cable reel so that its length can be adjusted. Using this adjustment mechanism, the initial attitudes of the modular units were aligned with a BOSCH self-levelling crossline laser; therefore, gravitational loads to the modular interfaces were minimised as the modular units were assembled. A vibration exciter was also suspended in the middle of the frame to apply loads to the test structure.

### 2.2. The Modular Units and Joints

As shown in Figure 1, the assembly consisted of three identical aluminium plates as mock-up modular units, each 2 m long, 0.5 m wide, and 2 mm thick. Preliminary theoretical predictions of the first five natural frequencies and mode shapes were performed using the transverse vibration equation of an equal-section beam [18]:

$$f_i = \frac{\lambda_i^2}{2\pi l^2}\sqrt{\frac{EI}{\rho S}}, i = 1, 2, 3, 4, 5, \tag{1}$$

$$Y_i(x) = \cos\frac{\lambda_i}{l}x + \cosh\frac{\lambda_i}{l}x + \left[\frac{\cos\lambda_i - \cosh\lambda_i}{\sinh\lambda_i - \sin\lambda_i}\right](\sin\frac{\lambda_i}{l}x + \sinh\frac{\lambda_i}{l}x), i = 1, 2, 3, 4, 5, \tag{2}$$

where $\lambda$ is the natural frequency parameter ($\lambda_{1-5} = \{4.730, 7.853, 10.996, 14.137, 17.279\}$), $l$ denotes the length of the beam, $E$ stands for the Young Modulus, $I$ represents the second moment of inertia of the beam section, $\rho$ is the density of the beam, and $S$ denotes the cross-sectional area of the beam. Using Equations (1) and (2), the theoretical predictions of the first five natural frequencies are listed in Table 1, and the corresponding mode shapes are illustrated in Figure 2.

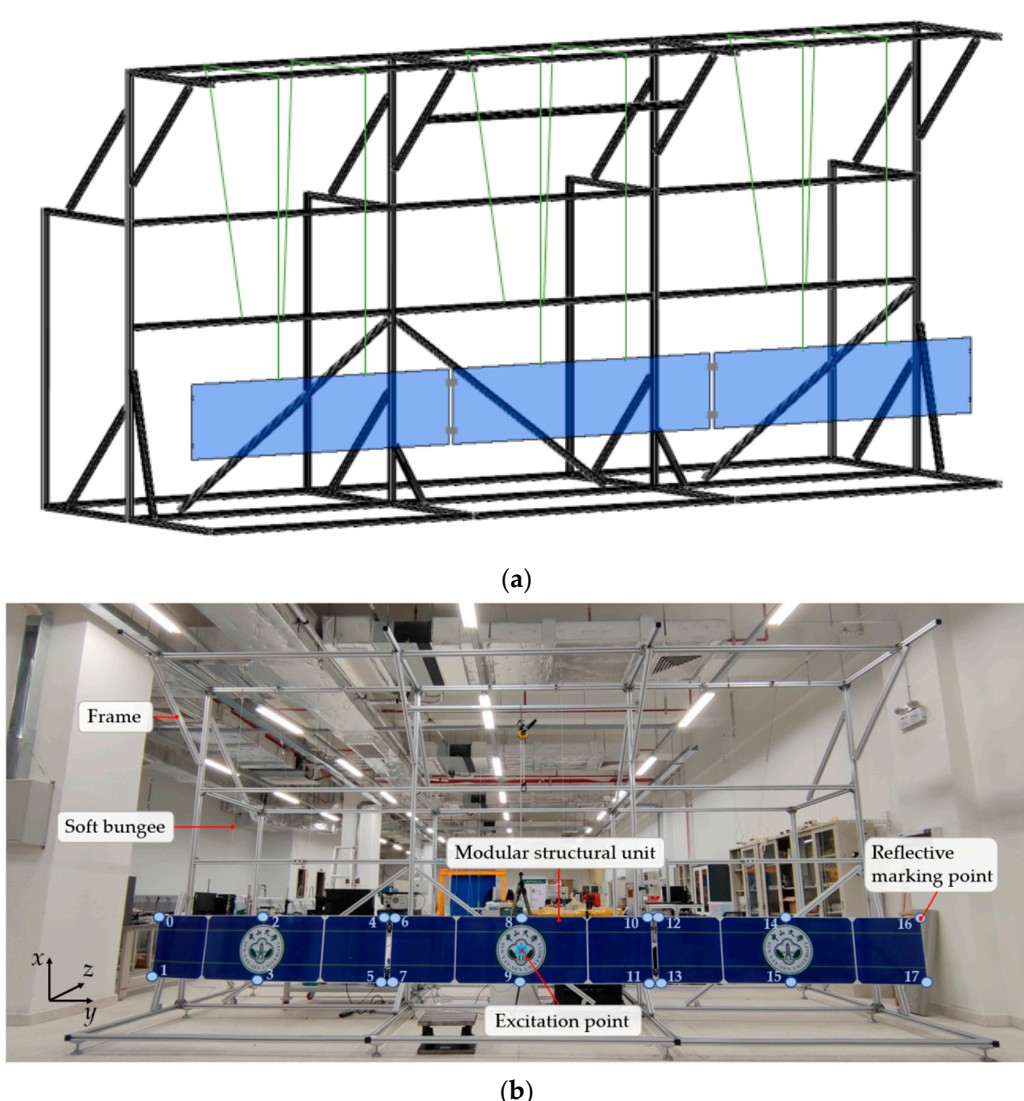

(**a**)

(**b**)

**Figure 1.** The suspension setup: (**a**) a CAD model and (**b**) a front-view picture, where locations of reflective markers are highlighted.

**Table 1.** Theoretical predictions of the first five natural frequencies of the assembly.

| Mode Order | Mode Shape | Theoretically Predicted Natural Frequency (Hz) |
|:---:|:---:|:---:|
| 1 | First bending mode | 0.29 |
| 2 | Second bending mode | 0.81 |
| 3 | Thirdbending mode | 1.59 |
| 4 | Forth bending mode | 2.63 |
| 5 | Fifth bending mode | 3.94 |

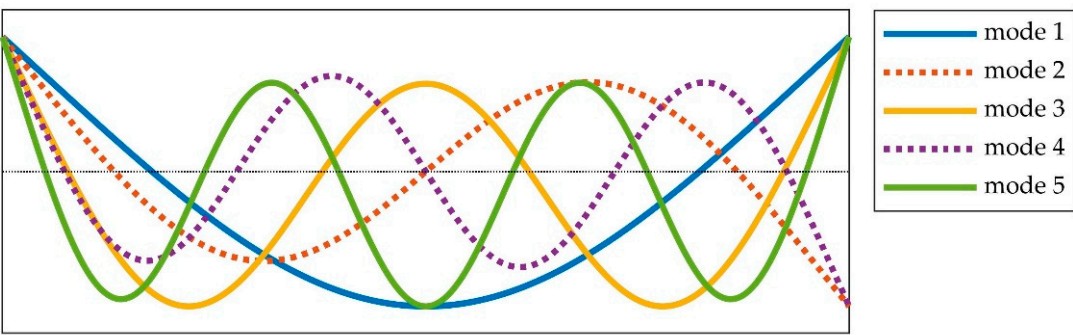

**Figure 2.** Theoretical predictions of the first five mode shapes of the assembly.

As shown in Figure 3, the three mock-up structural units were assembled using four quick-attachment quick-release joints, each weighing 53.6 g. Each joint contains two aluminium parts and a side latch, only requiring a simple push and pinch to attach or release.

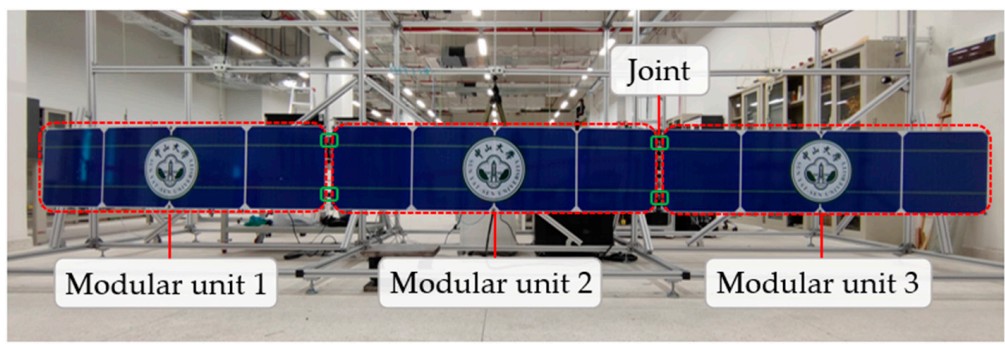

(**a**)

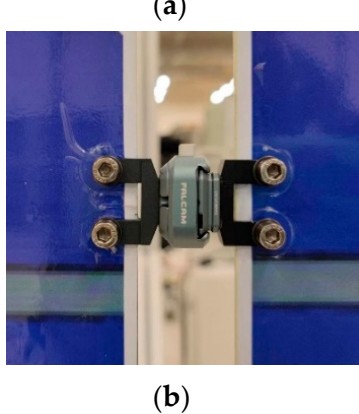

(**b**)

**Figure 3.** (**a**) Shows the modular units and (**b**) presents a zoomed-in view of the joint.

### 2.3. The Vibration Control System

In the resonance decay test, a vibration control system was used for force appropriation. As shown in Figure 4, a signal generator (Tektronix AFG2021) was used to provide the sinusoidal force command. The signal was amplified by a HEA-200C amplifier and then transmitted to an electromagnetic vibration exciter (DONGHUA DH40200). The force applied to the structure was measured by a force sensor (Kistler 3A105). Figure 4b shows that a capacitive tri-axial accelerometer (Kistler 8396A) with highly accurate DC measurement capability was also attached near the excitation point. As shown in the figure, it was glued to the structural unit, and its cable was secured with tape. The applied force and the acceleration at the driving point were then sampled with a multichannel data acquisition system (DONGHUA DH8303) in order to adjust the phase lags.

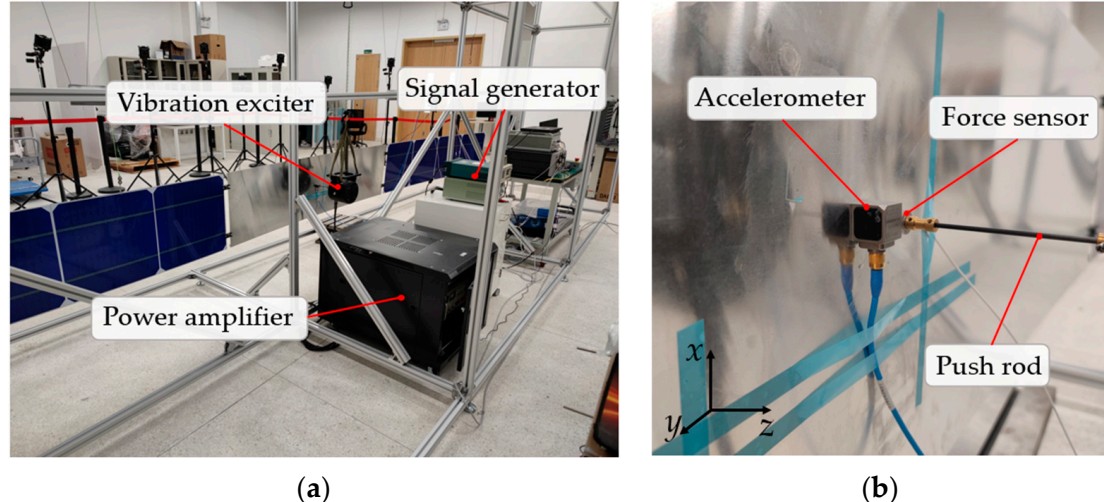

(**a**)　　　　　　　　　　　　　　　　　　　　　(**b**)

**Figure 4.** (**a**) The vibration control system and (**b**) the locations of the accelerometer and force sensor.

### 2.4. The Measurement System

Considering the very low natural frequencies of the modular assembly predicted in Table 1, a noncontact motion capture system (NOKOV) was established to perform 3D dynamic displacement measurements. As shown in Figure 1, a total of 18 passive markers were distributivity attached to the modular assembly as measurement targets, and 14 cameras were installed approx. 5 m away from the structure (see Figure 5). The NOKOV system tracked these markers simultaneously by infrared light and calculated the 3D coordinates of each marker with images from at least two cameras.

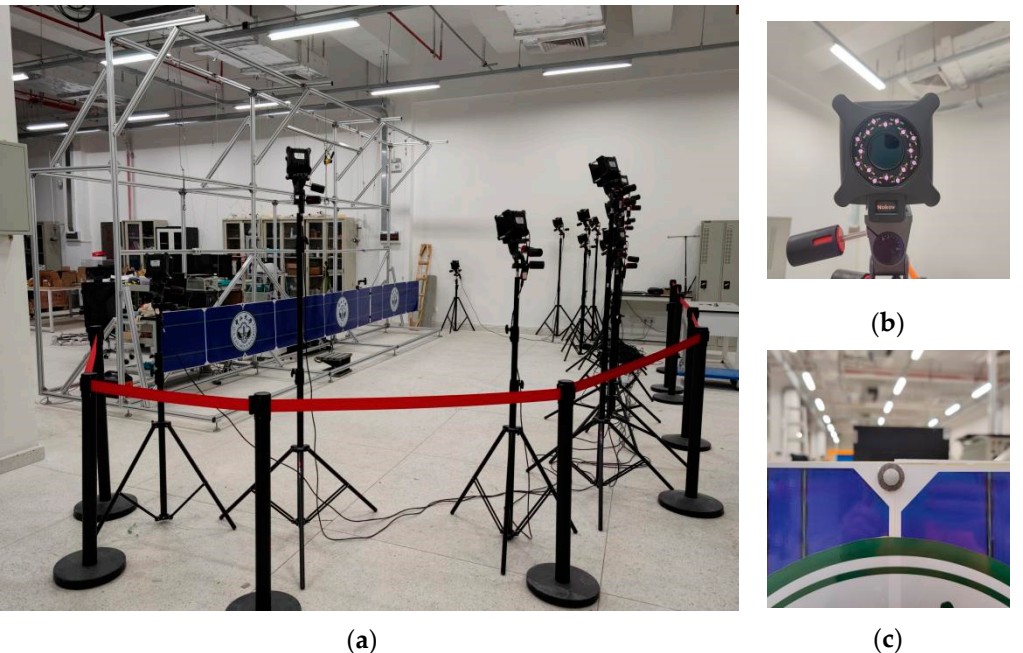

(**a**)　　　　　　　　　　　　　　　　　　　　　(**c**)

**Figure 5.** The measurement system: (**a**) overview of the noncontact motion capture system, (**b**) camera lens, and (**c**) reflective marking points attached to the test structure.

### 3. Quantification Procedure and Test Results

This section details the proposed quantification procedure for the dynamic testing of a modular assembly. Figure 6 shows a flowchart of the proposed procedure, which includes a linear modal survey test and a nonlinear resonance decay test. The detailed test steps and results will be introduced via the LSA demonstrator in the following subsections.

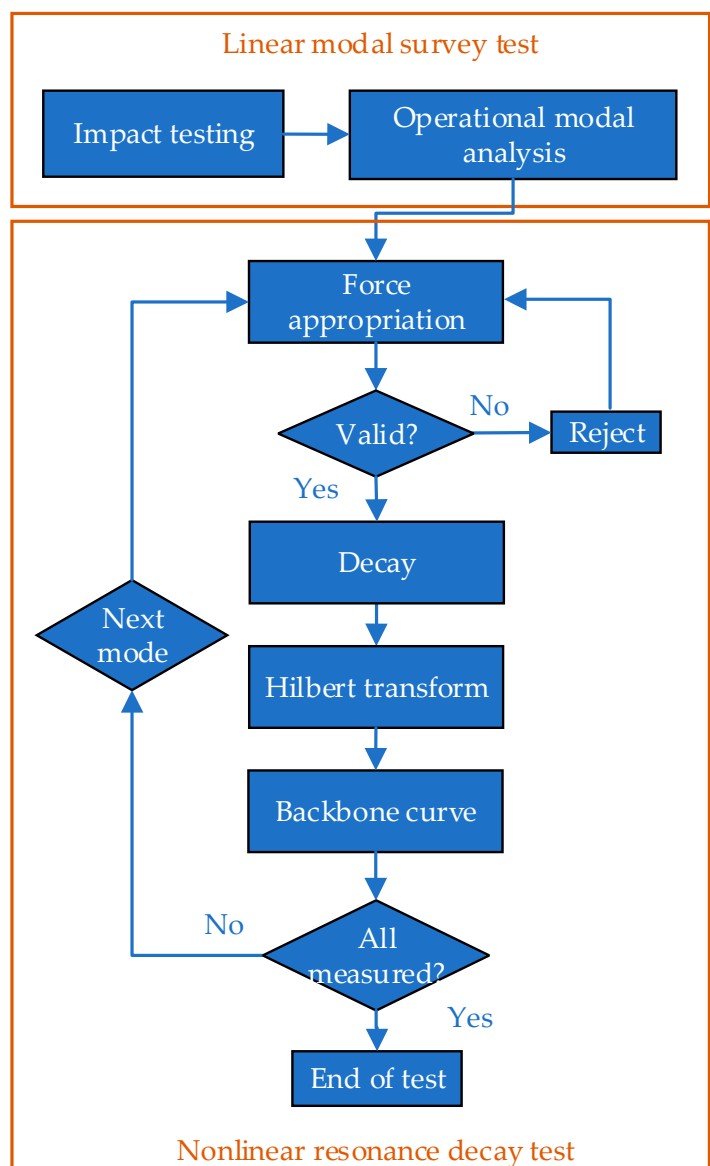

**Figure 6.** A flowchart of the proposed quantification procedure for a modular assembly.

### 3.1. Linear Modal Survey Test

Firstly, the linear normal modes of the modular assembly were identified using impact testing. Note that the impact force was not measured since it could not be synchronously sampled with the NOKOV motion capture system in the current test setup. Therefore, the response-only operational modal analysis (OMA) was applied to the test data measured by the motion capture system [19]. More specifically, the time-domain Stochastic Subspace Identification (SSI) algorithm [20] was used to identify modal parameters. Interested readers are referred to studies [19,20] for more details about the algorithm.

In the test of the modular assembly, the sampling rate of the motion capture system was set to 90 Hz, and the acquisition duration was 60 s. Figure 7 shows the stabilisation diagram estimated using the SSI algorithm, where five modes were clearly observed as the fitting model order reached 15. Table 2 lists the identified modal frequencies and damping ratios, and Figure 8 shows the corresponding modal shapes. Note also that the damping ratio of the first mode was not stable, so it was not shown in the table.

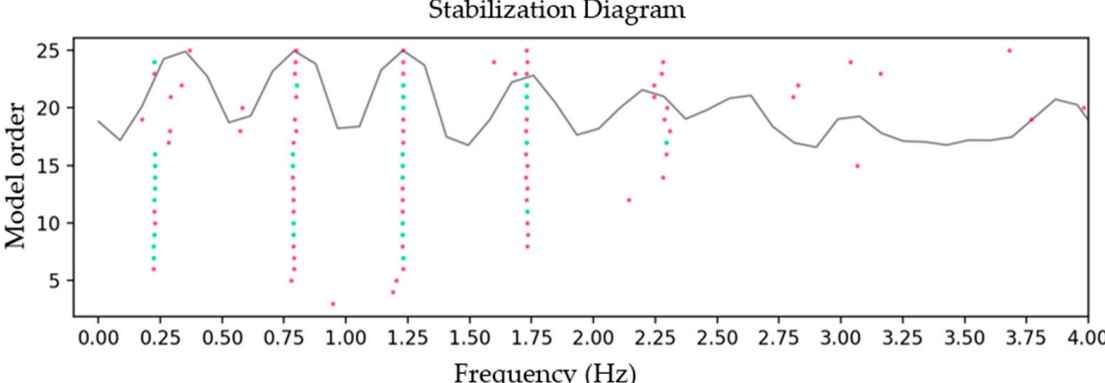

**Figure 7.** Stabilisation diagram with the following parameters: $\Delta\omega = 1\%$, $\Delta\zeta = 5\%$ and MAC = 0.98.

**Table 2.** Identified linear modal parameters using operational modal analysis.

| Mode Order | Mode Shape | Natural Frequency (Hz) | Modal Damping Ratio |
|:---:|:---:|:---:|:---:|
| 1 | First bending mode | 0.23 | |
| 2 | Second bending mode | 0.80 | 1.9% |
| 3 | Third bending mode | 1.23 | 1.1% |
| 4 | First sway mode | 1.73 | 1.1% |
| 5 | First torsion mode | 2.22 | 2.5% |

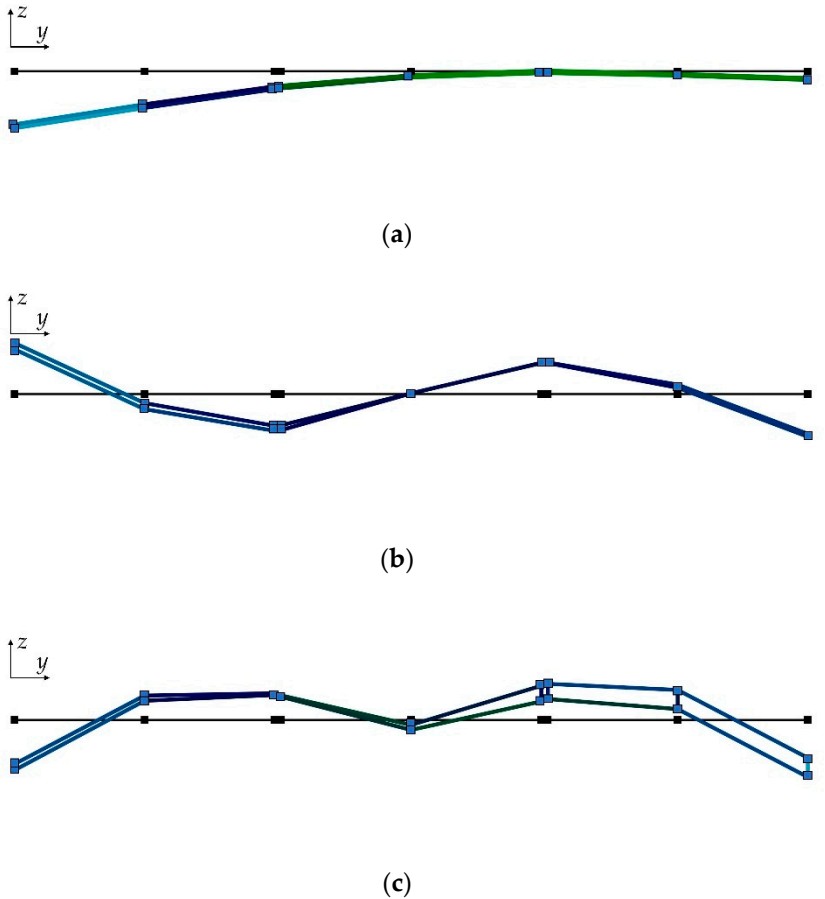

(**a**)

(**b**)

(**c**)

**Figure 8.** *Cont.*

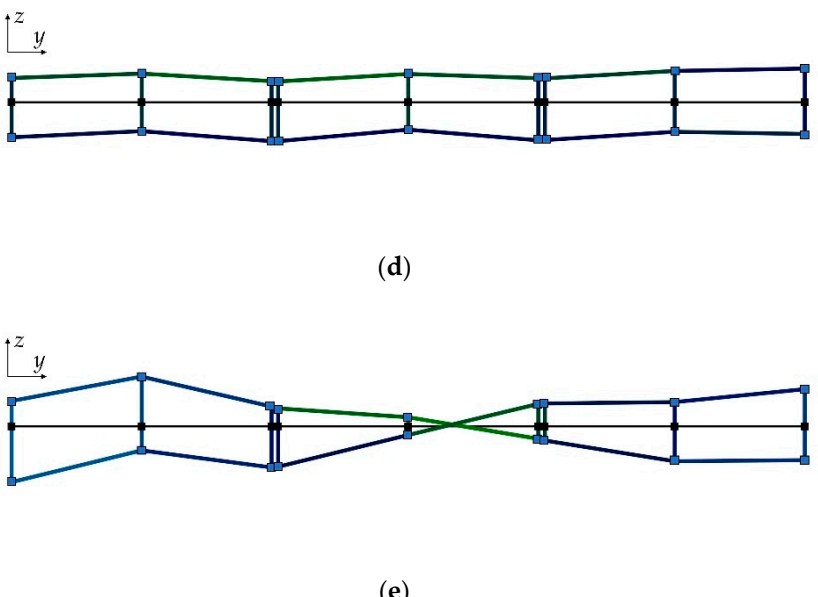

**(d)**

**(e)**

**Figure 8.** Identified mode shapes using operational modal analysis (top view): (**a**) mode 1, (**b**) mode 2, (**c**) mode 3, (**d**) mode 4, and (**e**) mode 5.

Comparing the first three identified modes (bending modes listed in Table 2) to the theoretical predictions using the beam theory (listed in Table 1), it was found that the natural frequencies of the real-life assembly were close but lower than the predicted values. This is reasonable since the adjacent modules were jointed only at two points (see Figure 3) while they were firmly line-connected in the continuous beam assumption.

*3.2. Nonlinear Resonance Decay Test*

In the proposed quantification procedure, the nonlinearities and uncertainties of the assembly dynamics are described using backbone curves, which are experimentally estimated using the resonance decay method [21]. In the method, a single structural resonance is appropriated using a sinusoidal force [22], and then the input force is removed to allow free vibrations of the modal responses to be measured. This method was initially developed for the identification of nonproportional damping of linear structures [21], and it was subsequently combined with the restoring force surface method [23] and the backbone curve estimation [24] to perform nonlinear system identification. It has the advantage of extracting the entire backbone curve of a nonlinear mode by performing a single resonance decay test, which is time efficient compared with nonlinear FRF-based methods [14] for highly flexible structures.

In the test of the modular assembly, the structural resonance was manually appropriated by adjusting the forcing frequency while monitoring the phase lags between the force and z-directional accelerations (see Figure 4). Until the phase lag was seen as 0° or 180°, the input force signal was removed to allow the structure to decay freely; the backbone curve of each mode was then estimated from the ringdown data using the Hilbert transform [25]. Note that this testing technique should be applied to each mode of interest, as shown in Figure 6.

Figure 9 shows the force appropriation results for mode 3 and mode 4 of the assembly, where the phase lags were seen as approx. 180° and 0°, respectively. It was also found that a proper excitation of mode 1 and mode 2 of the assembly was beyond the capability of the current vibration exciter. To apply loads to these two modes with ultralow frequencies (below 1 Hz), new exciters with longer strokes are required, which is beyond the scope of the current paper.

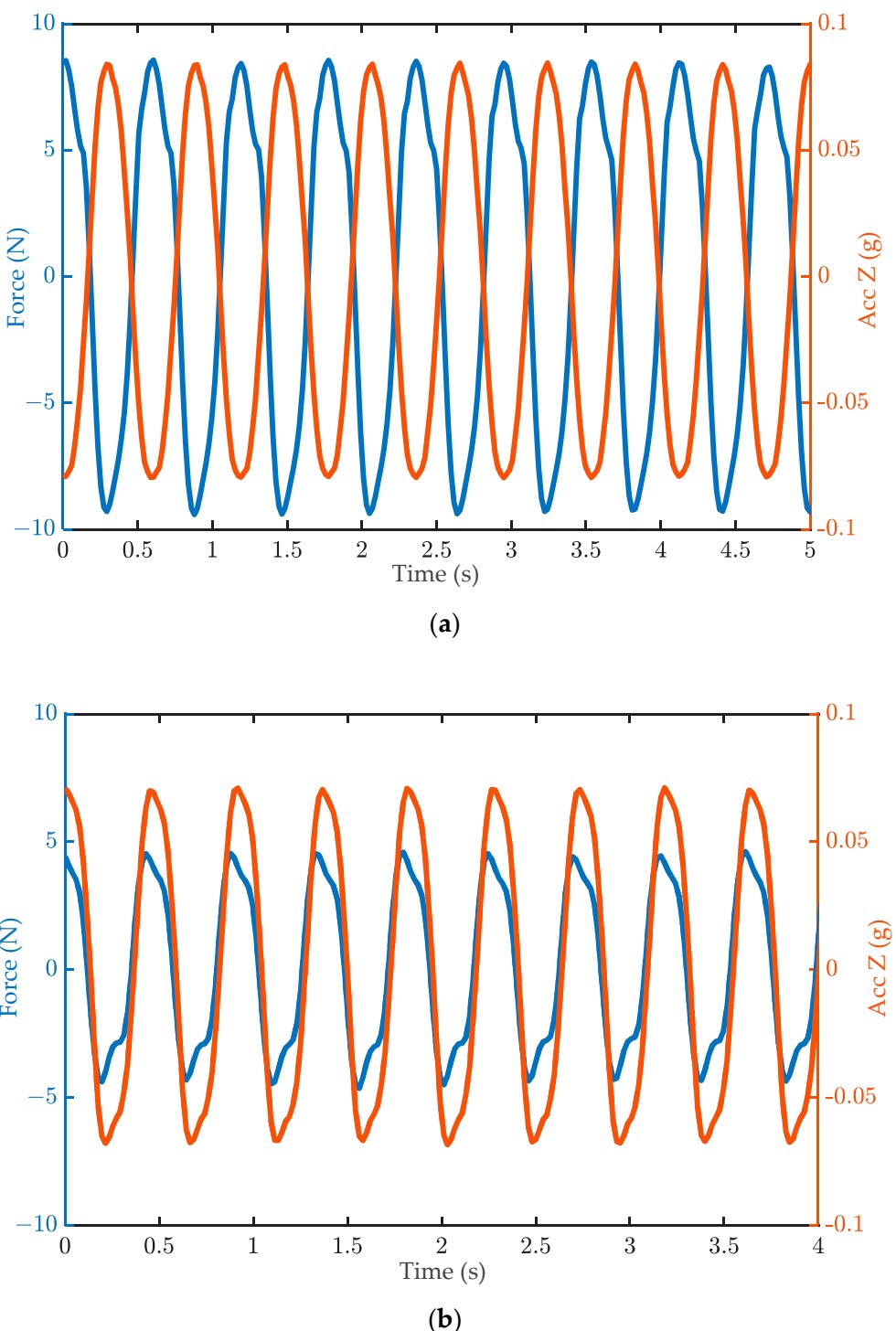

**Figure 9.** Force appropriation results for structural resonances: (**a**) a phase lag of 180° for mode 3 and (**b**) a phase lag of 0° for mode 4.

As soon as the phase resonance was appropriated with satisfactory accuracy (i.e., the phase lags between the acceleration and the input force were seen as approx. 180° or 0°), the input force signal was turned off, and the transient vibrations of the structure measured. The sampling rate of the vibration control system was set to 51.2 Hz, and a total of 80 s signal was recorded. The instantaneous frequency and damping ratios,

which were amplitude-dependent for a nonlinear structure, were then calculated from the ringdown data [25]:

$$\widetilde{\omega}_n^2(t) = \dot{\psi}^2 - \frac{\ddot{A}}{A} + \frac{2\dot{A}^2}{A^2} + \frac{\dot{A}\ddot{\psi}}{A\dot{\psi}}, \tag{3}$$

$$\widetilde{\zeta}(t) = \left(-\frac{\dot{A}}{A} - \frac{\ddot{\psi}}{2\dot{\psi}}\right)/\widetilde{\omega}_n(t), \tag{4}$$

where $A(t)$ and $\psi(t)$ are instantaneous estimates of the amplitude and phase of the vibration signals using the Hilbert transform [25].

One source of uncertainty is the variability of initial decay conditions. As such, the technique was performed three times for the same assembly. Figure 10 shows the test results of mode 3. It can be seen that, with the decrease of vibration amplitudes, the natural frequency gradually increased from 1.24 Hz to 1.25 Hz, while the damping ratio decreased from 1.2% to approx. 0.8%. Figure 11 illustrates the test results of mode 4, where a similar increase in natural frequencies was observed. Moreover, the damping ratios also decreased with the vibration amplitude. It was found that reasonably good repeatability was achieved for different ringdowns, despite the Hilbert transform giving fluctuated estimations of frequencies and damping ratios, which is already known as a limitation of the transform [24].

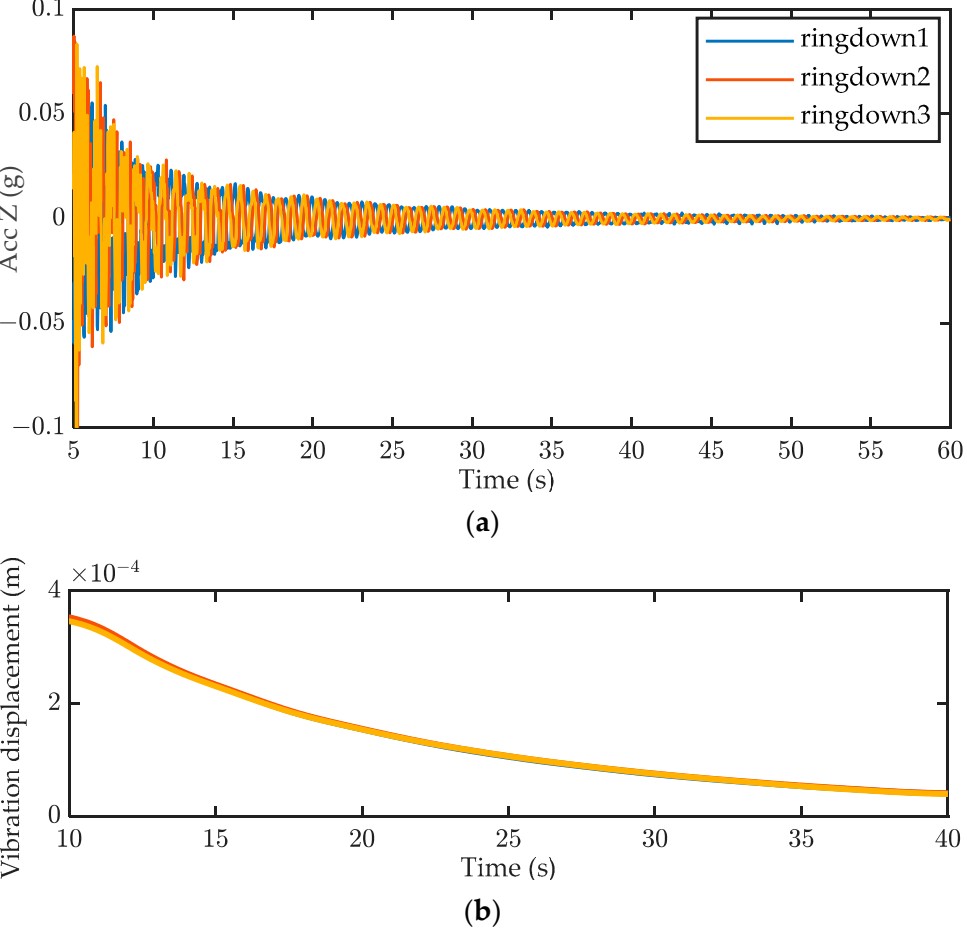

**Figure 10.** *Cont.*

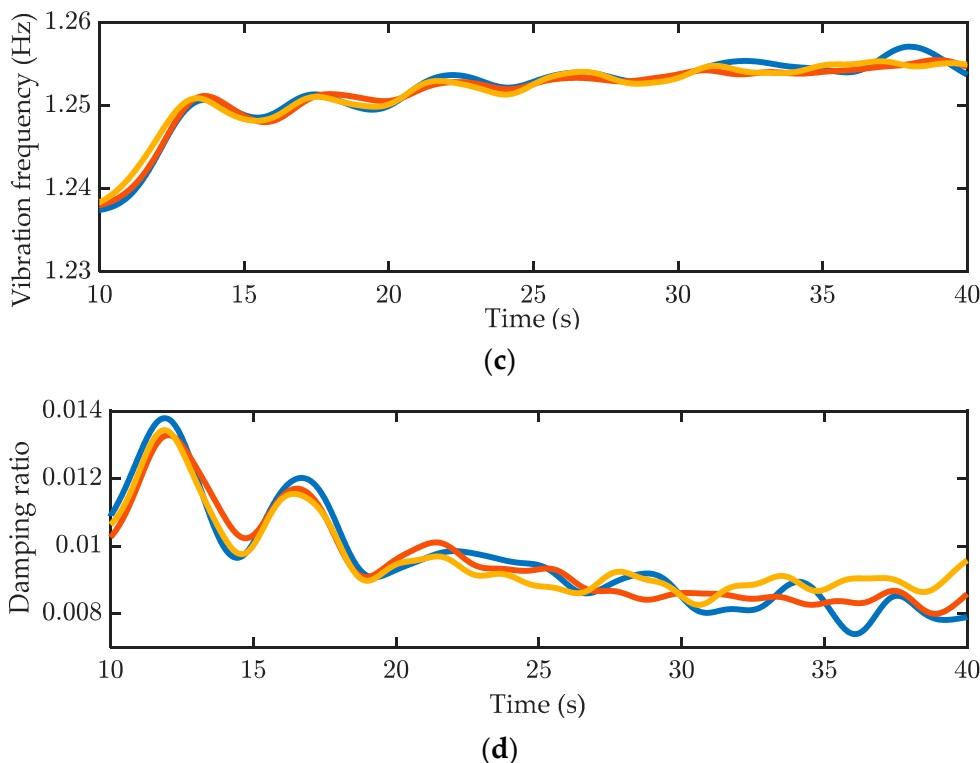

**Figure 10.** Resonance decay of mode 3: (**a**) ringdown data and estimations of (**b**) instantaneous displacement, (**c**) frequency and (**d**) damping.

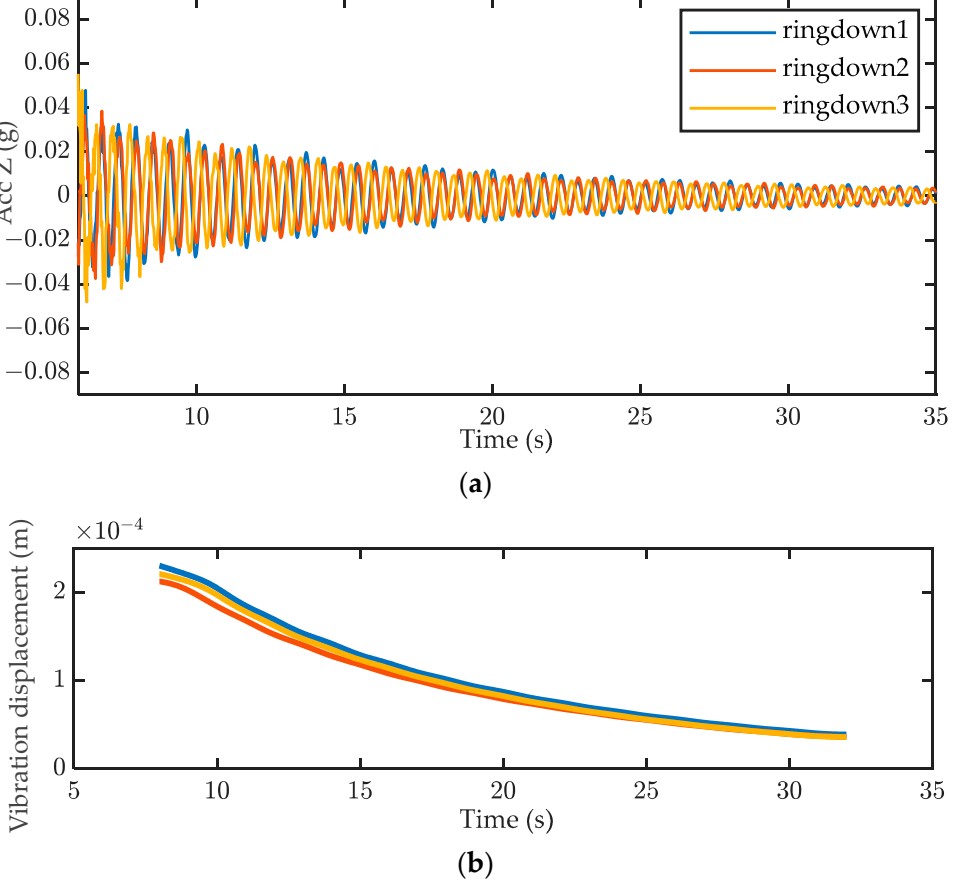

**Figure 11.** *Cont.*

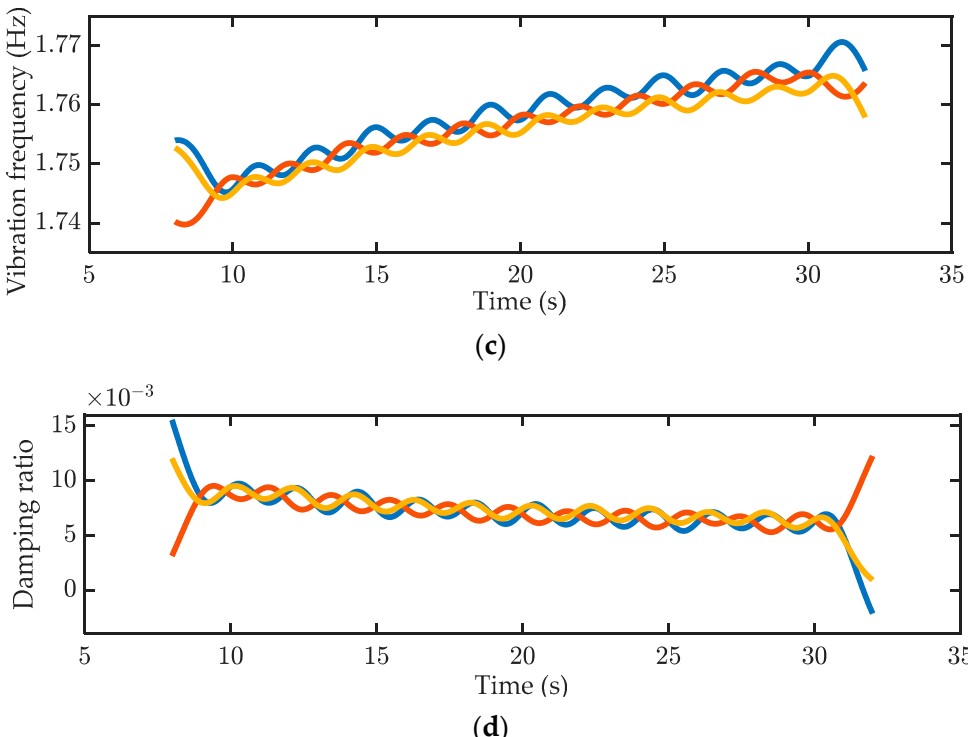

**Figure 11.** Resonance decay of mode 4: (**a**) ringdown data and estimations of (**b**) instantaneous displacement, (**c**) frequency, and (**d**) damping.

Another source of uncertainty is the variability associated with operations during the assembling process. Each time the structure is assembled, the clamping force of the joint varies, and the contact state changes accordingly. As such, the assembling process is an important source of uncertainty in the assembled structure. To quantify this, the structure was assembled and disassembled three times; the resonance decay test was performed for the modes of interest and for each assembly. Figure 12 shows the estimated backbone curves of mode 3 for the same test structure assembled three times, where the amplitude-dependent natural frequencies and damping ratios are illustrated. Only a trivial softening effect within a deviation of only 0.02 Hz can be seen from the backbone curve, illustrated in Figure 12a. In contrast, clear trends were found in the damping curve shown in Figure 12b, where the damping ratio increased from 0.8% to 1.2% with the vibration amplitude. Meanwhile, substantial uncertainties were also observed. Figure 13 depicts the estimated backbone curves for mode 4, where slight weak softening effects were observed, whilst the damping ratios increased rapidly from 0.6% to 1.0% with the vibration amplitudes, shown in Figure 13b.

It is interesting to compare the results obtained from the resonance decay test with those estimated in the impact test. Because of a trivial stiffness nonlinearity, the impact test and operational modal analysis provided very accurate estimations of the natural frequencies of the assembly. In addition, it detected lower-frequency modes that cannot be easily excited in the resonance decay test, in which a vibration exciter must be employed. The damping of the assembly, in contrast, showed substantial and inherent nonlinear behaviours, requiring the nonlinear resonance decay tests to quantify its trends.

Experimental applications to the LSA demonstrator have shown that the proposed procedure is time-efficient in testing, and the estimated backbone curves can describe uncertainties of the nonlinear dynamics of the assembly with good accuracy.

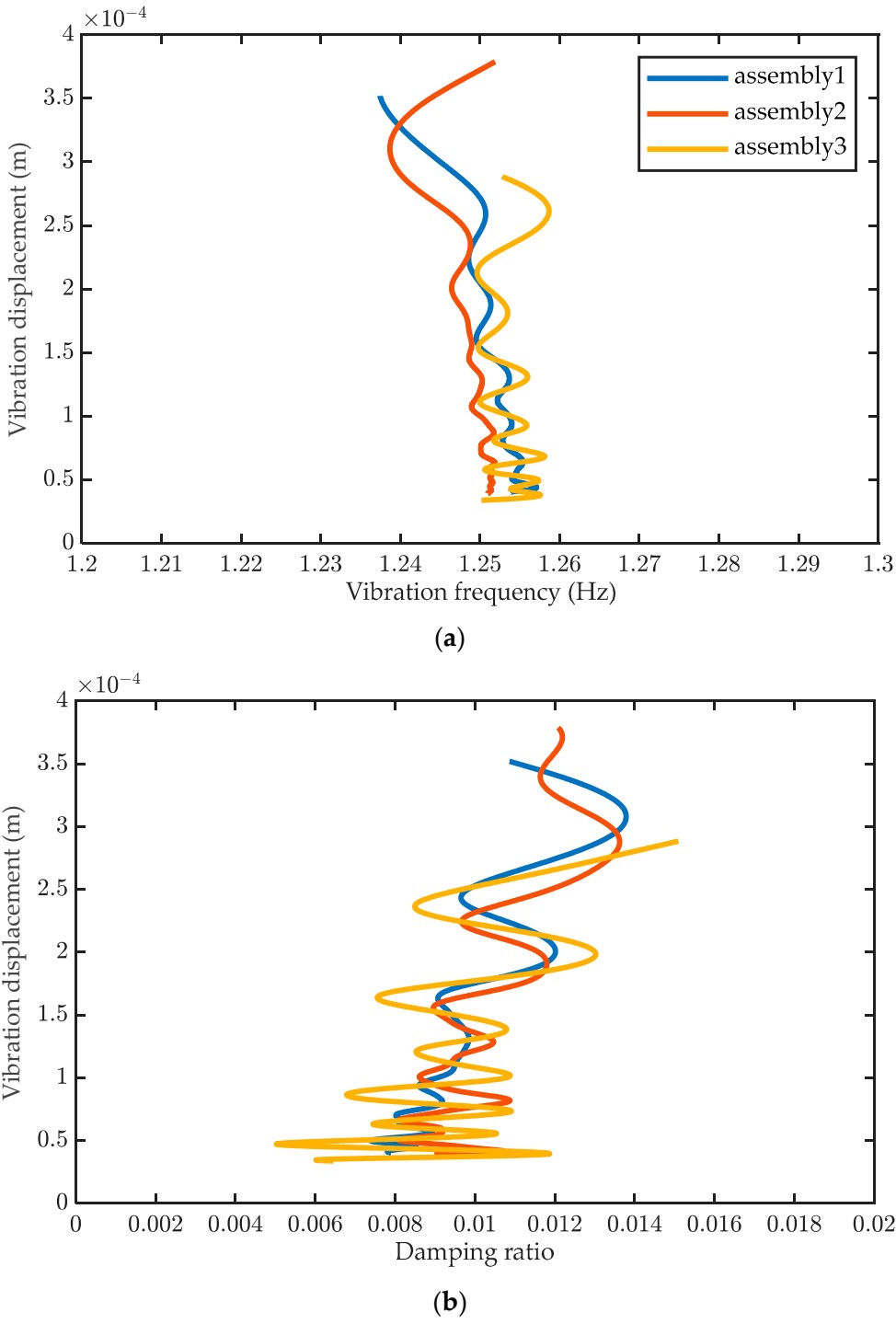

**Figure 12.** Backbone curves of mode 3: (**a**) natural frequency and (**b**) damping ratio.

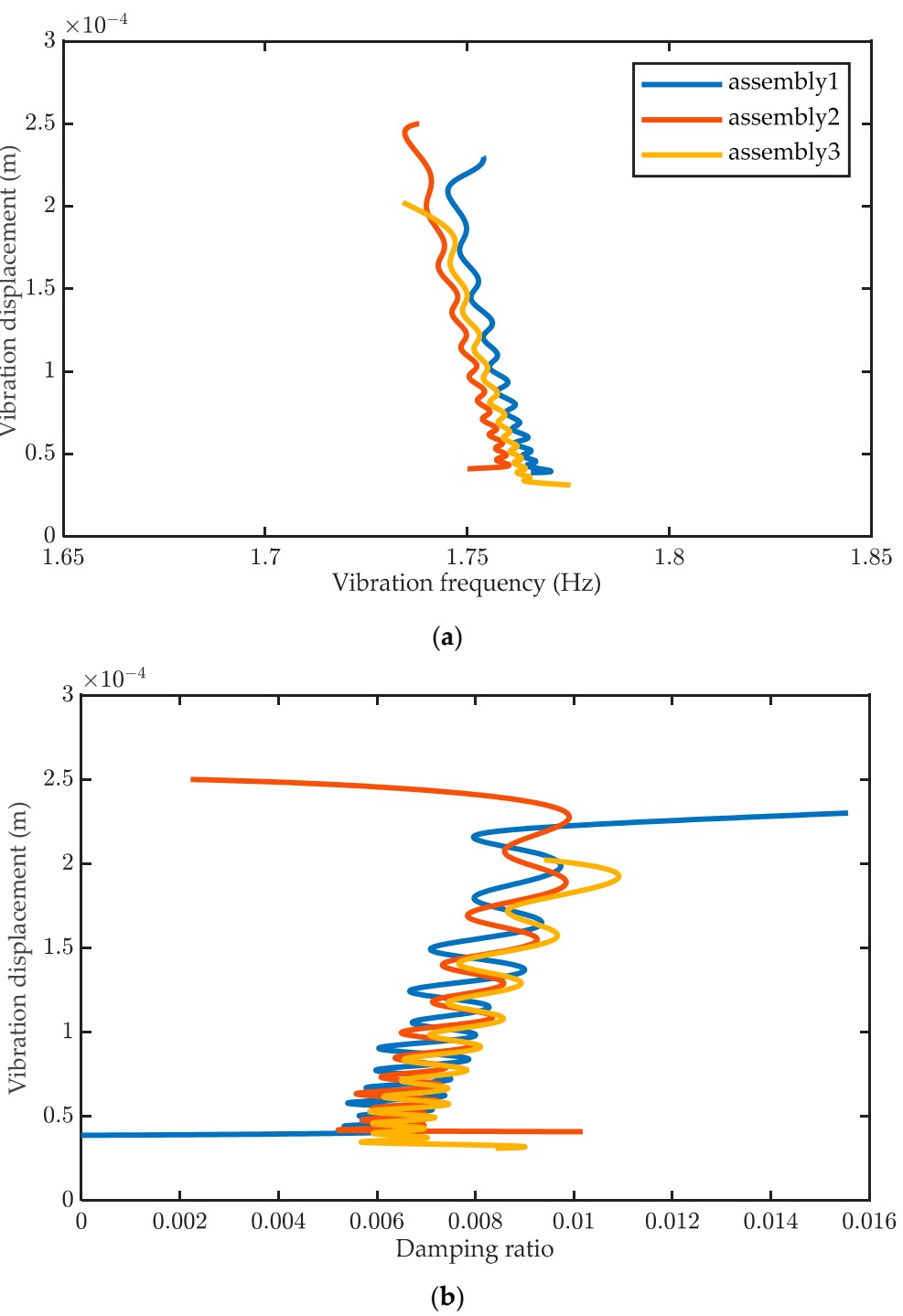

**Figure 13.** Backbone curves of mode 4: (**a**) natural frequency and (**b**) damping ratio.

## 4. Conclusions

The presented paper introduces a demonstrator of Large Structure Assembly developed to investigate advanced vibration testing techniques for highly flexible, large-scale modular assemblies. Using the LSA demonstrator, a new quantification procedure for uncertainties and nonlinearities of structural dynamics is proposed and validated. It first uses impact testing and operational modal analysis to identify the underlying linear modal properties of the assembly. Then, the resonance decay method is applied for each mode of interest, where backbone curves are estimated to describe the amplitude-dependent natural

frequencies and damping ratios. In addition, multiple repeats of the tests and multiple assembling processes are used to quantify the uncertainties of the dynamics.

Through the GVT testing campaign of a real-life modular assembly, the following conclusions are obtained:

(1) The proposed quantitation procedure uses backbone curves as measurement targets, which can be efficiently measured using the nonlinear resonance decay method. It is also shown that backbone curves can also accurately describe the uncertainties and nonlinearities of the assembly dynamics;

(2) The modular assembly of the LSA demonstrator showed very weak stiffness nonlinearities (less than 2% softening was observed in natural frequencies), while the damping ratios varied as much as 66.7% with the vibration amplitudes. It certainly highlights the necessity of a dedicated quantification procedure for the damping nonlinearities, which cannot be identified using conventional techniques such as impact testing.

Hardware upgrade to use a vibration exciter with longer stroke is feasible. Further discussions of the suspension system on the dynamics of the assembly may also be helpful. In addition, the proposed quantification procedure may have limitations in applying to structures with heavily damped modes.

**Author Contributions:** Conceptualisation, X.W.; methodology, C.L., Z.Z. and X.W.; validation, X.W.; formal analysis, C.L., Z.Z., Z.W. (Zhenyu Wang) and X.W.; writing—original draft preparation, C.L., Z.Z. and X.W.; writing—review and editing, J.J., Z.W. (Zhigang Wu) and X.W.; visualisation, C.L., Z.Z., Z.W. (Zhenyu Wang) and X.W.; supervision, X.W.; funding acquisition, Z.W. (Zhigang Wu) and X.W. All authors have read and agreed to the published version of the manuscript.

**Funding:** This research was funded by the Shenzhen Science and Technology Program (Grant No. RCYX20210706092137055), the National Natural Science Foundation of China (Grants No. 52005522, 12072378, 11872381), and the Fundamental Research Funds for the Central Universities, Sun Yat-sen University (Grant No. 22qntd0703).

**Data Availability Statement:** The data in this paper is available upon request.

**Acknowledgments:** X.W. would like to gratefully acknowledge the support from the Shenzhen Science and Technology Program (Grant No. RCYX20210706092137055), the National Natural Science Foundation of China (Grants No. 52005522 and 12072378) and the Fundamental Research Funds for the Central Universities, Sun Yat-sen University (Grant No. 22qntd0703). Z.W. thanks the National Natural Science Foundation of China (Grant No. 11872381) for financial support.

**Conflicts of Interest:** The authors declare no conflict of interest.

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
