# Peer review of "Quantifying Uncertainties in Nonlinear Dynamics of a Modular Assembly Using the Resonance Decay Method"

_actuators, doi:10.3390/act11120350_

Round 1

Reviewer 1 Report

The work is interesting. Minor comments in the appendix. It should be explained how the accelerometer was attached to the vibrating substrate. The stiffness and damping of the accelerometer-substrate connection can significantly interfere with the measurement results. The literature should be supplemented with the indicated references.

Reviewer 2 Report

Modular assembling is a candidate approach to build large spacecraft in space. This manuscript introduced a demonstrator for Large Structure Assembly and proposed a novel quantification procedure for such structures, and it can be accepted for publication. However, a few minor issues have to be addressed.

(1) Page 13-14, figures 10-11. Please clarify the filters used to plot the ringdown data, since it may affect the quality of the results.

(2) The authors are recommended to add a few limitations of the proposed technique at the end of conclusion.

Reviewer 3 Report

The paper discloses a methodology for vibration testi g of large, assembled structures with low natural frequencies.

The paper is interesting and well written.

Some suggestions to improve it:

1) Introduction is not clear in the part where the work is described. It would be better to spend a few additional sentences for this

2) it would be interesting to have a  better description of the connection between the plates and which parameters can change each time assembly is prepared
